# From Discrete Tokens to High-Fidelity Audio Using Multi-Band Diffusion

**Robin San Roman**◇,♠    **Yossi Adi**◇,♣    **Antoine Deleforge**♠    **Romain Serizel**♠

**Gabriel Synnaeve**◇                **Alexandre Defossez**◇

◇: FAIR Team, Meta
♠: Universite de Lorraine, CNRS, Inria, LORIA, Nancy, France
♣: The Hebrew University of Jerusalem

## Abstract

Deep generative models can generate high-fidelity audio conditioned on various types of representations (e.g., mel-spectrograms, Mel-frequency Cepstral Coefficients (MFCC)). Recently, such models have been used to synthesize audio waveforms conditioned on highly compressed representations. Although such methods produce impressive results, they are prone to generate audible artifacts when the conditioning is flawed or imperfect. An alternative modeling approach is to use diffusion models. However, these have mainly been used as speech vocoders (i.e., conditioned on mel-spectrograms) or generating relatively low sampling rate signals. In this work, we propose a high-fidelity multi-band diffusion-based framework that generates any type of audio modality (e.g., speech, music, environmental sounds) from low-bitrate discrete representations. At equal bit rate, the proposed approach outperforms state-of-the-art generative techniques in terms of perceptual quality. Training and evaluation code are available on the facebookresearch/audiocraft github project. Samples are available on the following link.

## 1 Introduction

Neural-based vocoders have become the dominant approach for speech synthesis due to their ability to produce high-quality samples [Tan et al., 2021]. These models are built upon recent advancements in neural network architectures such as WaveNet [Oord et al., 2016] and MelGAN [Kumar et al., 2019], and have shown impressive results in generating speech with natural-sounding intonation and timbre.

In parallel, Self-Supervised Learning (SSL) applied to speech and audio data [Hsu et al., 2021, van den Oord et al., 2019] have led to rich contextual representations that contain more than lexical content, e.g., emotion and prosody information [Kharitonov et al., 2022, Kreuk et al., 2022a]. Generating waveform audio from such representations is hence a new topic of interest [Liu et al., 2019, Polyak et al., 2021, Huang et al., 2022]. This is often performed in a two stage training pipeline. First, learn audio representations using SSL objectives, then, decode the speech using Generative Adversarial Networks (GAN) approach such as the HiFi GAN model [Kong et al., 2020a]. Even though these methods perform well, they are known to be unstable, difficult to train and prone to add audible artifacts to the output waveform.

Compression models [Zeghidour et al., 2021, Défossez et al., 2022] can also be considered as SSL models that use the reconstruction loss as a way to learn meaningful representations of the data. Unlike models described before, compression models are trained in an end-to-end fashion, while learning

37th Conference on Neural Information Processing Systems (NeurIPS 2023).

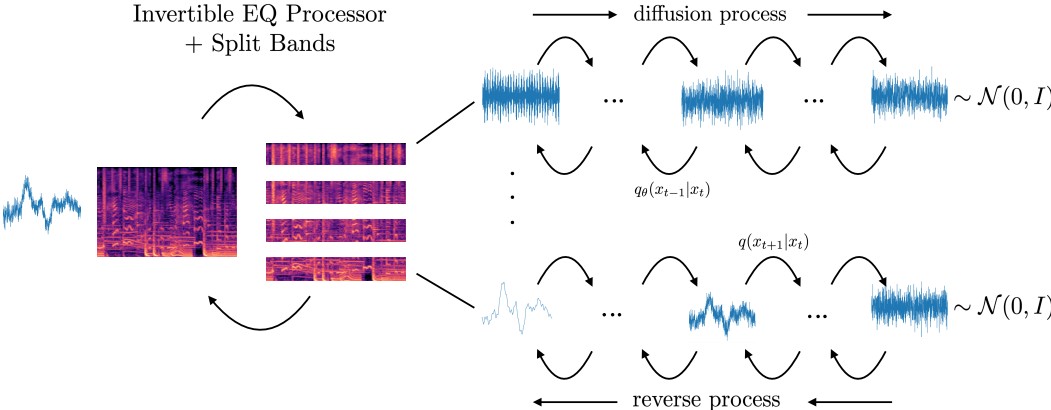

Figure 1: MULTI-BAND DIFFUSION process (resp. reverse process). The first step consists of a reversible operation (EQ Processor) that normalizes the energy within frequency bands to resemble that of a standard Gaussian noise. The audio is then filtered into non-overlapping bands. Each band has its own diffusion process using a specifically tuned version of the proposed *power noise schedule*.

both audio representation (often a discrete one) and synthesis, and can model a large variety of audio domains. They are optimized using complex combinations of specifically engineered objectives including spectrogram and feature matching, as well as multiple adversarial losses [Défossez et al., 2022]. Even though they have impressive performance compared to standard audio codecs, e.g., Opus [Valin et al., 2012], they tend to add noticeable artefacts when used at very low bit rates (e.g. metallic voices, distortion) that are often blatantly out of distribution.

After model optimization the learned representation can also be used for different audio modeling tasks. Kreuk et al. [2022b] presented a textually guided general audio generation. Wang et al. [2023] proposed a zero shot text to speech approach. Agostinelli et al. [2023] demonstrated how such representation can be used for text-to-music generation, while Hsu et al. [2022] followed a similar modeling approach for silent video to speech generation.

In this work, we present MULTI-BAND DIFFUSION (MBD), a novel diffusion-based method. The proposed approach can generate high-fidelity samples in the waveform domain of general audio, may it be speech, music, environmental sounds, etc. from discrete compressed representations. We evaluate the proposed approach considering both objective metrics and human studies. As we demonstrate empirically, such an approach can be applied to a wide variety of tasks and audio domains to replace the traditional GAN based decoders. The results indicate that the proposed method outperforms the evaluated baselines by a significant margin.

**Our Contributions:** We present a novel diffusion based model for general audio synthesis. The proposed method is based on: (i) a **band-specific diffusion model** that independently processes different frequency bands, allowing for less accumulative entangled errors; (ii) a **frequency equalizer (EQ) processor** that reduces the discrepancy between the prior Gaussian distribution and the data distribution in different frequency bands; and (iii) A novel **power noise scheduler** designed for audio data with rich harmonic content. We conduct extensive evaluations considering both objective metrics and human study, demonstrating the efficiency of the proposed approach over state-of-the-art methods, considering both GAN and diffusion based approaches.

## 2   Related work

Neural audio synthesis has been originally performed with sample level autoregressive models such as Wavenet [Oord et al., 2016]. This type of architecture is notoriously slow and difficult to train. Speech synthesis is one of the dominant area of audio synthesis research. Vocoders are models designed to generate waveform speech from low level phonetic or acoustic features. Different approaches, often

conditioned on mel-spectrograms, have been explored for this task, including GAN-based methods such as HiFi-GAN [Kong et al., 2020a, Kumar et al., 2019]. Polyak et al. [2021] used HiFi-GAN on other types of conditioning. This method generate speech conditioned on low bit rate representations learned from self-supervised methods such as HuBERT [Hsu et al., 2021] VQ-VAE [van Niekerk et al., 2020] or CPC [van den Oord et al., 2019] together with the fundamental frequency $f_0$ and a speaker embedding. By using only a few centroids for clustering, the speech content representation becomes largely disentangled from the speaker and the fundamental frequency (f0), enabling controllable speech generation.

Diffusion-based vocoders are inspired by the recent success of diffusion for image generation [Ho et al., 2020, Saharia et al., 2022, Dhariwal and Nichol, 2021, Ramesh et al., 2022]. Kong et al. [2020b] introduced Diffwave, a diffusion-based vocoders, that applies the vanilla diffusion equations to waveform audio. Compared with the adversarial approach, diffusion offers a simpler L2 Loss objective, and stable training. PriorGrad [Lee et al., 2021] is an extension of Diffwave that uses non standard Gaussian noise in the diffusion process. The authors extract the energy of the conditioning mel-spectrogram and use it to adapt the prior noise distribution to the target speech. Wavegrad [Chen et al., 2020] is similar but uses conditioning on continuous noise levels instead of discrete ones. This allows the model to perform the sampling using any noise schedule with a single training. Takahashi et al. [2023] look at singing voices, which is a more complex distribution than standard read speech due to wider spectrum, and increased diversity. Inspired by super-resolution cascaded techniques from image diffusion [Ho et al., 2022], they used hierarchical models. The first diffusion model synthesises at a low sampling rate while later ones, conditioned on the output of their predecessor, perform upsampling. This process can yield high-quality, high-resolution audio samples. Recent work [Pascual et al., 2022] applies diffusion to generating full band audio at high sampling rate, although the proposed methods allows for unconditional generation, and flexible style transfer, it remains limited to a narrow range of audio modalities.

Most diffusion models that sample data from complex high dimensional distributions use upsampling frameworks [Huang et al., 2023, Takahashi et al., 2023]. This type of cascaded models are achieving good performance but they are based on series of diffusion processes conditioned on the output of the previous and thus can not be performed in parallel. In vision, some efforts have been invested in simplifying diffusion pipelines. SimpleDiffusion [Hoogeboom et al., 2023] presents a framework that matches results of cascading diffusion models using a single model. The model architecture and training objective are adapted to focus on low-frequency content while keeping high quality textures. To the best of our knowledge, this type of idea has not been ported to audio processing as of yet.

Finally, our work offers an alternative to the decoder of adversarial neural audio codecs such as SoundStream [Zeghidour et al., 2021] and EnCodec [Défossez et al., 2022], which consist in an encoder, a quantizer, and a decoder, and are trained with combination of losses including discriminators, spectrogram matching, feature matching, and waveform distances. Our diffusion based decoder is compatible, but offers higher quality generation as measured by subjective evaluations.

## 3 Method

### 3.1 Background

Following Ho et al. [2020], we consider a diffusion process given by a Markov chain $q$ where Gaussian noise is gradually added to corrupt a clean data point $x_0$ until a random variable $x_T$ close to the standard Gaussian noise is obtained. The probability of the full process is given by

$$q(x_{0:T}|x_0) = \prod_{t=1}^{T} q(x_t|x_{t-1}), \tag{1}$$

where $q(x_t|x_{t-1}) \sim \mathcal{N}(\sqrt{1-\beta_t}x_{t-1}, \beta_t I)$ and $(\beta_t)_{0 \leq t \leq T}$ is usually referred to as the noise schedule. One can efficiently sample any step of the Markov chain $t$ with

$$x_t = \sqrt{\bar{\alpha}_t}x_0 + \sqrt{1-\bar{\alpha}_t}\varepsilon, \tag{2}$$

where $\bar{\alpha}_t = \prod_{s=0}^{t}(1-\beta_s)$ is called the noise level and $\varepsilon \sim \mathcal{N}(0, I)$. Denoising Diffusion Probabilistic Models (DDPM) aims at going from prior noise $x_T$ to the clean data point $x_0$ through the

reverse process

$$p(x_{T:0}) = p(x_T) \prod_{t=1}^{T} p_\theta(x_{t-1}|x_t), \tag{3}$$

where $p_\theta(x_t|x_{t+1})$ is a learned distribution that reverses the diffusion chain $q(x_{t+1}|x_t)$ and $p(x_T)$ is the so-called *prior* distribution that is not learned. Under the ideal noise schedule, one can see from eq. (2) that the prior distribution can be approximated by $\mathcal{N}(0, I)$.

Ho et al. [2020] show that the distribution $p_\theta(x_{t-1}|x_t)$ can be expressed as $\mathcal{N}(\mu_\theta(x_t, t), \sigma_t I)$ where $\mu_\theta$ can be reparameterized as follow:

$$\mu_\theta(x_t, t) = \frac{1}{\sqrt{1 - \beta_t}} \left( x_t - \frac{\beta_t}{\sqrt{1 - \bar{\alpha}_t}} \varepsilon_\theta(x_t, t) \right). \tag{4}$$

This reparametrization allows to train a neural network $\varepsilon_\theta$ to predict the noise in the corrupted data point $x_t$. To train this neural network, one can use the simple objective given by Ho et al. [2020] that consists in sampling $x_t$ using eq. (2) and optimizing the following L2 loss:

$$\mathcal{L} = \mathbf{E}_{x_0 \sim d(x_0), \varepsilon \sim \mathcal{N}(0,I), t \sim \mathcal{U}\{1,..,T\}} \left( ||\varepsilon - \varepsilon_\theta(\sqrt{\bar{\alpha}_t} x_0 + \sqrt{1 - \bar{\alpha}_t} \varepsilon, t)||^2 \right). \tag{5}$$

With such a model, one can reverse the diffusion process iteratively using the following equation:

$$x_{t-1} = \frac{1}{\sqrt{1 - \beta_t}} \left( x_t - \frac{\beta_t}{\sqrt{1 - \bar{\alpha}_t}} \varepsilon_\theta(x_t, t) \right) + \sqrt{\sigma_t} \varepsilon, \tag{6}$$

where $\sigma$ is a parameter that should be chosen between $\tilde{\beta}_t = (1 - \bar{\alpha}_{t-1})/(1 - \bar{\alpha}_t)\beta_t$ and $\beta_t$ [Ho et al., 2020]. In our experiments we always use $\sigma_t = \tilde{\beta}_t$.

## 3.2 Multi-Band Diffusion

The MULTI-BAND DIFFUSION method is based on three main components: (i) Frequency EQ processor; (ii) Scheduler tuning; and (iii) Band-specific training, which we now describe.

**Frequency Eq. Processor** The mathematical theory of diffusion processes allows them to sample from any kind of distribution, regardless of its nature. However, in practice, training a diffusion network for multiple audio modalities in the waveform domain is an open challenge. We make the assumption that the balance of energy levels across different frequency bands in both the prior Gaussian distribution and the target distribution is important to obtain an efficient sampling mechanism.

A white Gaussian noise signal has equal energy over all frequencies. However natural sounds such as speech and music do not follow the same distribution Schnupp et al. [2011], i.e. music signals tend to have similar energy level among frequency bands that are exponentially larger. For signals of the same scale, white noise has overwhelmingly more energy in the high frequencies than common audio signals, especially at higher sample rate (see Fig. 2). Thus during the diffusion process, high frequency content will disappear sooner than the low frequency counterpart. Similarly, during the reverse process, the addition of noise given by (6) will have more impact over the high frequencies.

To resolve this issue, we normalize the energy of the clean signal, denoted as $x_0$, across multiple frequency bands. We split $x_0$ into $B$ components $b_i$, with a cascade of band pass filters equally spaced in mel-scale. Given the filtered band $b_i$ of an audio signal, we normalize as follow,

$$\hat{b}_i = b_i \cdot \left( \frac{\sigma_i^\epsilon}{\sigma_i^d} \right)^\rho, \tag{7}$$

where $\sigma_i^\epsilon$ and $\sigma_i^d$ denote the energies in the band $i$ for standard Gaussian noise and for the signals in the dataset, respectively. The parameter $\rho$ controls to what extend we align the energy levels. For $\rho = 0$ the processor does not do any rebalancing and $\rho = 1$ corresponds to matching exactly the target energy. Given that speech signals often have no content in the high frequency bands, we compute the parameters $\sigma_i^d$ over the music domain to avoid instabilities in (7).

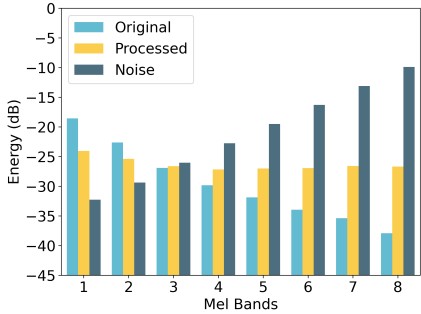

Figure 2: Standard deviation in 8 mel scale frequency bands (from lows to highs). For data from our dataset (Original), Equalized data (Processed) and for standard Gaussian Noise (Noise).

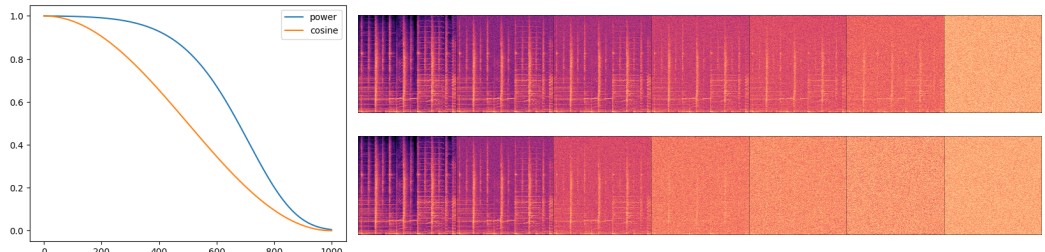

Figure 3: Left curves depict a comparison of the noise level $\bar{\alpha}$ along the diffusion process for cosine schedule and our power schedule. Right figure presents spectrograms along diffusion process. The top row is our power schedule and the bottom row follows cosine schedule.

**Scheduler Tuning.** The noise schedule is known to entail an important set of hyperparameters in diffusion models, and to play a critical role in the final quality of the generation [Karras et al., 2022].

A common approach when generating raw waveform is using either linear or cosine schedules [Nichol and Dhariwal, 2021]. Such schedulers performs good for read speech where the frequency spectrum is not wide or for low-sampling-rate generation followed by cascaded models that iteratively upsample the signal. In preliminary experiments, we found such schedules performs poorly when generating signals at high sampling rate. Hence, we argue that one should prefer a more drastic schedule. We propose to use *p-power schedules*, defined as:

$$\beta_t = \left( \sqrt[p]{\beta_0} + \frac{t}{T}(\sqrt[p]{\beta_T} - \sqrt[p]{\beta_0}) \right)^p, \tag{8}$$

where the variance of the injected noise at the first and last step ($\beta_0$ and $\beta_T$) are hyperparameters. One could assume that, since the noise schedule used at generation time can be chosen after training, it is unnecessary to tune the training noise schedule and only focus on the choice of subset $S$. As evoked by Chen et al. [2020], in practice, the training noise schedule is a crucial element of diffusion models. Since the train-time steps are sampled uniformly, the training noise schedule parameterizes the sampling distribution of the noise level $\sqrt{\bar{\alpha}}$. As seen in Fig. 3, using the proposed power schedule will results in sampling most of the training examples very small amount of noise (i.e. very high $\bar{\alpha}$).

We noticed that for a time step $t$ close to $T$, i.e. at the end of the diffusion process, the model estimate $\epsilon_\theta(x_t)$ of the noise is often worse than simply using $x_t$ itself. We hypothesize this is due to the limited precision when training the model. In that regime, the model can advantageously be replaced by the identity function, which is equivalent to skipping those timesteps entirely. We thus choose the $\beta_t$ values such that $\sqrt{1 - \alpha_t}$ is large enough to avoid this phenomenon.

**Band-Specific Training.** Similarly to the findings in image diffusion models [Song et al., 2020], audio diffusion models first generate low frequencies and then address high frequencies during the final stages of the reverse process. Unlike images where high frequencies are only locally connected, audio data contains complex entanglements of spectrogram content across both time and frequency

[Schnupp et al., 2011]. As a result, training a diffusion model on full-band audio data would always provide the ground truth low frequencies when generating high frequencies. It ends up amplifying the errors committed at the beginning of the generation when unrolling the reverse process.

Following that, we proposed training each frequency band independently, denoted as MULTI-BAND DIFFUSION . Through preliminary experiments we found such an approach resulted in significant improvements in the perceptual quality of the samples. Interestingly, dividing the frequency band along model channels did not yield the same quality improvements. This observation supports our intuition that, by not providing the model with previously generated content (lower frequencies) during training, the model can avoid accumulating errors during sampling.

# 4    Experimental Setup

## 4.1    Model & Hyperparameters

**Overview.** Our approach serves as a replacement for EnCodec's decoder. This approach offers the advantage of flexibility and compatibility. It allows one to switch between the original decoder and the proposed diffusion decoder depending on the required trade-off between quality and generation time.

**Architecture.**  Similarly to Chen et al. [2020], Kong et al. [2020b], Lee et al. [2021], we use a fully convolutional symmetrical U-net network [Ronneberger et al., 2015] with an alternation of two residual blocks [Défossez et al., 2021] and downsampling (resp. upsampling in the decoder) convolutional blocks of stride 4. The input audio conditioning is incorporated in the bottleneck of the network whereas the timestep $t$ is embedded through a learned lookup table and added at every layer. According to the recommendation of Hoogeboom et al. [2023], it is advisable to allocate additional computational resources close to the bottleneck of the network when applying diffusion to high-dimensional data. Hence, we opted for a growth rate of 4. The weight of one model is 1 GB. A visual description of the model architecture can be seen in Fig. A.4 in the Appendix.

**Input Conditioning.** We use the latent representation of the publicly available EnCodec models at 24kHz [Défossez et al., 2022] which are frozen during training.  The embedding sequence is upsampled using linear interpolation to match the dimension of the UNet bottleneck.  In the experiments we include reconstructions using 1, 2 and 4 for EnCodec code books which correspond to bit rates of respectively 1.5kbps, 3kbps and 6kbps, when using multiple code books the embedding used is simply the average of the different code books.

**Schedule.** We trained our diffusion models using our proposed power schedule with power $p = 7.5$, $\beta_0 = 1.0e{-}5$ and $\beta_T = 2.9e{-}2$. Although we use very few diffusion steps (20) at generation time, we observed that it is beneficial to use many steps at training time (1000). First, it increases the versatility of the model since one can sample using any subset of steps $S \subseteq \{1, \dots 1000\}$. Second, it allows the model to be trained on a more diverse range of noise levels $\sqrt{\bar{\alpha}_t}$. In the experiment section we always use the simplest time steps sub sampling i.e. $S = \{i * \frac{1000}{N}, i \in \{0, 1, ..., N\}\}$ where $N$ is the number of sampling steps (20 if not precised).

**Frequency EQ processor.** In the experiments we use a band processor that uses 8 mel scale frequency bands with $\rho = 0.4$. We compute the values of the bands $\sigma_i^d$ on an internal music dataset.

**Band Splitting.** As described in 3 we use separate diffusion processes. In this work we always use a split of 4 frequency bands equally space in mel-scale using `julius` [1] Those bands are not related to the processor bands. The 4 models share the same hyperparameters and schedule. All models take the same ENCODEC tokens as conditioning input.

**Training.** We train our models using Adam optimizer with batch size 128 and a learning rate of 1e-4. It takes around 2 days on 4 Nvidia V100 with 16 GB to train one of the 4 models.

**Computational cost and model size.** Diffusion model sampling has an intrinsic cost that is due to the number of model passes that are required for generation. We provide in Table A.7 the details for time consumption and number of parameters of MULTI-BAND DIFFUSION .

---

[1] https://github.com/adefossez/julius

### 4.2 Datasets

We train on a diverse set of domains and data. We use speech from the train set of Common Voice 7.0 (9096 hours) [Ardila et al., 2019] together with the DNS challenge 4 (2425 hours) [Dubey et al., 2022]. For music, we use the MTG-Jamendo dataset (919h) [Bogdanov et al., 2019]. For the environmental sound we use FSD50K (108 hours) [Fonseca et al., 2021] and AudioSet (4989 hours) [Gemmeke et al., 2017]. We used AudioSet only for the research that is described in the publication and for the benefit of replicability. For evaluation, we also use samples from an internal music dataset.

### 4.3 Evaluation Metrics

**Human evaluation.** For the human study we follow the MUSHRA protocol [Series, 2014], using a hidden reference and a low anchor. Annotators were recruited using a crowd-sourcing platform, in which they were asked to rate the perceptual quality of the provided samples in a range between 1 to 100. We randomly select 50 samples of 5 seconds from each category of the the test set and force at least 10 annotations per samples. To filter noisy annotations and outliers we remove annotators who rate the reference recordings less then 90 in at least 20% of the cases, or rate the low-anchor recording above 80 more than 50% of the time.

**Objective metrics.** We use two automatic evaluation functions. The first one is the standard ViSQOL [Chinen et al., 2020] metric. [2]. The second one is a novel metric we introduce to measure the fidelity of the mel-spectrogram of the reconstructed signal compared with the ground truth across multiple frequency bands. Let us take a reference waveform signal $x \in \mathbb{R}^T$ and a reconstructed signal $\hat{x} \in \mathbb{R}^T$. We normalize $x$ to have unit variance, and use the same scaling factor for $\hat{x}$. We take the mel-spectrogram of both, computed over the power spectrum with $M$ mels, and a hop-length $H$, e.g.,

$$z = \mathrm{mel}\left[ \frac{x}{\epsilon + \sqrt{\langle x^2 \rangle}} \right], \qquad \text{and} \qquad \hat{z} = \mathrm{mel}\left[ \frac{\hat{x}}{\epsilon + \sqrt{\langle x^2 \rangle}} \right], \qquad (9)$$

with $z, \hat{z} \in \mathbb{R}^{F \times T/H}$. We compute the mel-spectrogram distortion $\delta = z - \hat{z}$. Finally for each time step $t$ and frequency bin $f$, we can compute a Signal-To-Noise ratio. In order to avoid numerical instabilities, and also not let the metric be overly impacted by near zero values in the ground truth mel-spectrogram, we clamp the SNR value between $-25dB$, $+25dB$, considering that any distortion lower than -25 dB would have a limited impact on perception, and that beyond +25 dB, all distortions would be equally bad. Indeed, due to the limited precision used in the computation and training of a neural network, it is virtually impossible to output a perfectly null level of energy in any given frequency band, although such empty bands could happen in real signals. Finally we get,

$$s = \mathrm{clamp}\left[ 10 \cdot (\log_{10}(z) - \log_{10}(\delta))., -25\mathrm{dB}, +25\mathrm{dB} \right]. \qquad (10)$$

We then average over the time steps, and split the mel-scale bands into 3 equally spaced in mel-scale. We report each band as Mel-SNR-L (low frequencies), Mel-SNR-M (mid frequencies), and Mel-SNR-H (high frequencies). Finally we also report the average over all 3 bands as Mel-SNR-A. At 24 kHz, we use a STFT over frames of 512 samples, with a hop length $H = 128$ and $N = 80$ mel bands.

## 5 Results

### 5.1 Multi modalities model

We first evaluate the performance of our diffusion method compared with ENCODEC on a compression task. Specifically we extract audio tokens from audio samples using the ENCODEC encoder and decode them using MULTI-BAND DIFFUSION and the original decoder.

We perform subjective evaluation on four subsets: 50 samples of clean speech from DNS, 50 samples of corrupted speech using DNS blended with samples from FSD50K, 50 music samples from Jamendo and 50 music samples from an internal music dataset. All speech samples are reverberated with probability 0.2 using room impulse responses provided in the DNS challenge. In Table 1, we present 3 subjective studies with different bit rate levels: 6kbps, 3kbps, and 1.5kbps. Note that scores should not be compared across the studies since ratings are done relatively to the other samples of the study. We include Opus [Valin et al., 2012] at 6kbps as a low anchor and the ground truth samples. Even

---

[2]We compute visqol with: `https://github.com/google/visqol` using the recommended recipes.

Table 1: Human evaluations (MUSHRA) scores for 24kHz audio. The mean and CI95 results are reported. The Opus low anchor and ground truth samples are consistent across all three studies, delimited by horizontal lines. The other methods used a bit rate of 6kbps for the top study on top, 3kbps for the middle one, and 1.5kbps for the bottom one. Higher scores indicate superior quality.

| Method | Speech | Music | Average |
|---|---|---|---|
| Reference | $93.86_{\pm0.014}$ | $92.93_{\pm0.021}$ | 93.40 |
| Opus | $61.14_{\pm0.094}$ | $34.24_{\pm0.147}$ | 47.69 |
| ENCODEC | $79.03_{\pm0.053}$ | $\mathbf{84.67_{\pm0.062}}$ | 81.85 |
| MBD (ours) | $\mathbf{84.68_{\pm0.036}}$ | $83.61_{\pm0.072}$ | $\mathbf{84.15}$ |
| Reference | $93.17_{\pm0.015}$ | $94.45_{\pm0.014}$ | 93.81 |
| Opus | $62.83_{\pm0.14}$ | $36.17_{\pm0.12}$ | 49.5 |
| ENCODEC | $78.51_{\pm0.078}$ | $85.55_{\pm0.045}$ | 82.03 |
| MBD (ours) | $\mathbf{84.42_{\pm0.042}}$ | $\mathbf{87.31_{\pm0.041}}$ | $\mathbf{85.87}$ |
| Reference | $94.65_{\pm0.012}$ | $94.71_{\pm0.012}$ | 94.78 |
| Opus | $44.65_{\pm0.057}$ | $38.33_{\pm0.081}$ | 41.49 |
| ENCODEC | $49.51_{\pm0.072}$ | $\mathbf{75.98_{\pm0.077}}$ | 62.75 |
| MBD (ours) | $\mathbf{65.83_{\pm0.056}}$ | $75.29_{\pm0.076}$ | $\mathbf{70.56}$ |

Table 2: Human evaluations (MUSHRA) scores for 24kHz audio. The mean and CI95 results are reported. The first part of the table reports different methods of ENCODEC tokens at 6kbps decoding while the second part adds other independent compression baselines at 6 kbps.

| Method | score |
|---|---|
| Ground Truth | $90.32_{\pm1.39}$ |
| MBD | $\mathbf{85.16}_{\pm0.93}$ |
| Encodec | $82.73_{\pm1.11}$ |
| PriorGrad | $65.16_{\pm2.2}$ |
| HifiGan | $82.5_{\pm1.25}$ |
| DAC | $84.44_{\pm1.14}$ |
| OPUS | $65_{\pm2.43}$ |

though the comparison with ENCODEC is done with different model sizes cf Table A.7, original paper Défossez et al. [2022] makes it clear that the number of parameters of the model is not a limiting factor of their method.

Multi-Band Diffusion outperform EnCodec on speech compression by a significant margin, up to 30% better, while being on part on music data. Averaging across modalities, our method outperforms EnCodec for all bit rates. Qualitatively, we observed that GAN-based methods have a tendency to introduce very sharp and straight harmonics that can lead to metallic artifacts. On the other hand, our diffusion method produces more blurred high-frequency content. We provide a number of spectrogram in the Supplementary Material, Section A.2.

In table 2, we compare our approach with other decoders baseline trained using the same condition and data as our model. Specifically we compare to HifiGAN Kong et al. [2020a] and PriorGrad Lee et al. [2021] using the hyper parameters proposed on their original papers.

The second part of table 2 adds comparisons to other end to end audio codecs that do not rely on ENCODEC. Specifically it adds the pretrained model of DAC Kumar et al. [2023] at 6kpbs which is a different audio codec at 24khz. We show ENCODEC + MULTI-BAND DIFFUSION is on part with DAC that uses a different quantized space. It is likely that training our MULTI-BAND DIFFUSION on the audio tokens of DAC would results in even higher audio quality.

Table 3: Objective and subjective metrics comparing the reconstruction performances of our model and ENCODEC across bit rates.

| Setting | ViSQOL (↑) | Mel-SNR-L (↑) | Mel-SNR-M (↑) | Mel-SNR-H (↑) | Mel-SNR-A (↑) |
|---|---|---|---|---|---|
| MBD @1.5kbps | 3.20 ±0.02 | 10.09 | 8 .03 | 8.26 | 8.79 |
| ENCODEC@1.5kbps | 3.33±0.02 | 9.61 | 10.8 | 13.37 | 11.36 |
| MBD 3.0 kbps | 3.47±0.02 | 11.65 | 8.91 | 8.69 | 9.75 |
| ENCODEC@3.0kbps | 3.64±0.02 | 11.42 | 11.97 | 14.34 | 12.55 |
| MBD @6.0 kbps | 3.67±0.02 | 13.33 | 9.85 | 9.26 | 10.81 |
| ENCODEC@6.0kbps | 3.92±0.02 | 13.19 | 12.91 | 15.21 | 13.75 |

Table 4: Comparing the reconstruction performances of our model at 6kbps.

| Setting | ViSQOL (↑) | Mel-SNR-L (↑) | Mel-SNR-M (↑) | Mel-SNR-H (↑) | Mel-SNR-A (↑) |
|---|---|---|---|---|---|
| MBD @6.0 kbps | **3.67**±0.02 | **13.33** | **9.85** | 9.26 | **10.81** |
| w-o Processor | 3.38±0.02 | 13.16 | 9.68 | 8.46 | 10.43 |
| Linear Schedule | 2.93±0.03 | 10.65 | 7.10 | 7.73 | 8.49 |
| Cosine Schedule | 3.29±0.03 | 12.88 | 9.60 | **9.59** | 10.69 |
| Single Band | 3.32±0.02 | 12.76 | 9.82 | 8.58 | 10.39 |

'

## 5.2 Ablations

In this section we use objective metrics to compare the reconstruction performances. We compute for every experiments the ViQOL score and Mel-SNR on 3 mel spec bands. Objective metrics are computed on the same 4 types of modalities as in section 5.1 using 150 samples per category. Even though those metrics seem to not correlate well with human evaluation across different model families (c.f. Tables 1 and 3) in our testing it was accurately measuring the improvements in quality resulting from small design changes. In Table 3, we compare the reconstruction performances of Multi-Band Diffusion and Encodec at different bit rates. It is notable that overall Encodec achieves better objective reconstruction metrics while being outperformed in subjective evaluations. We argue that such models are better in spectral distance metrics due to their specific training for content reconstruction. On the other hand, diffusion based methods do not use feature or spectrogram matching and tend to create samples that are more "in distribution" resulting in more natural audio. Diffusion based methods have more freedom to generate something that will be different from the original audio. They are optimized to keep maximize likelihood of their output with respect to the train dataset. The optimal method might be different depending on the purpose. However we claim that our MULTI-BAND DIFFUSION is preferable for most generative tasks based on generation in the codec space.

To evaluate the impact of our individual contributions we performed an ablation study that evaluates models in the exact same setting when removing one element introduced in this article.

According to the findings of our study, increasing the number of steps to 20 results in improved output quality. However, further increasing the number of steps shows diminishing returns (results available in the Appendix Table 4). In comparison to our approach utilizing four models, a single model performs less effectively. Despite employing a similar number of neural function estimations it has worse audio quality and worse scores in every metrics. By leveraging our processor to rebalance the frequency bands, we achieved a notable enhancement of 0.2 in ViSQOL scores. Additionally, our proposed schedule demonstrates a performance increase of 0.4 and 0.2 when compared to standard linear and cosine schedules Nichol and Dhariwal [2021]. Moreover, our proposed data processing technique also leads to a 0.2 increase in ViSQOL scores. The figures displayed in table 3 indicate that the high frequencies (Mel-SNR-H) are primarily affected by this processing technique.

## 5.3 Text to audio

Although our model alone cannot generate audio without conditioning, we show that when combined with a generative language model on the audio tokens, it provides substantial quality enhancements.

**Text to Speech.** Using language models on audio codecs has recently gained interest for Text to Speech. Methods such as VALL-E Wang et al. [2023] or SPEAR-TSS Kharitonov et al. [2023] achieved convincing results on this task. We claim that one can improve the quality of the final

Table 5: Human evaluations (MUSHRA) decoding token sequences from various methods.

| Method | Speech | Bark Singing Voices | Average | MusicGen Music |
|---|---|---|---|---|
| ENCODEC | $64.34_{\pm3.6}$ | $61.85_{\pm4.2}$ | 63.10 | $70.99_{\pm1.19}$ |
| MBD | $\mathbf{76.04}_{\pm2.9}$ | $\mathbf{73.67}_{\pm3.4}$ | **73.86** | $\mathbf{74.97}_{\pm1.94}$ |

audio by just switching to our MULTI-BAND DIFFUSION token decoder To test that claim we use the implementation and pretrained models from Bark[3] that are publicly available. Bark is composed of three transformer models. The initial model converts text input into high-level self-supervised audio tokens, while the second and third models sequentially process these tokens to produce Encodec tokens with two and eight codebooks, respectively. We used our trained diffusion models to decode the final token sequences. We generated 50 text prompts from Bark in all supported languages. We also include 50 prompts using the music note emoji as suggested in the official Github page to generate some singing voices. We removed from the subjective tests the samples for which the language model failed to generate any voice, in our experiments using pretrained bark this append for less than 5% of speech prompts and around 30% singing voice prompts. Table 5 presents the results, and we include the Encodec generation used in the original code base as a baseline.

**Text to Music.** There has been a significant advancement in the field of music generation using language modeling of audio tokens. Recently, this progress has been exemplified by MusicLM [Agostinelli et al., 2023] and MusicGen [Copet et al., 2023], which have greatly improved text-to-music generation. In order to demonstrate the versatility of our decoding approach, we utilized the open source version of MusicGen and trained a diffusion model conditioned with the tokens produced by its compression model. Our model is trained on the same dataset as the EnCodec model used by MusicGen, with a sampling rate of 32kHz. Additionally, we match the standard deviation of 16 mel scaled bands with the compression model output.

Notably, our method achieved a MUSHRA score improvement of +4 compared to standard MusicGen (see Table 5). Overall, the artifacts generated by the diffusion decoder are less pronounced. We find that in music containing complex elements, such as fast drum playing, the outputs from MULTI-BAND DIFFUSION are much clearer than the original ones.

## 6  Discussion

In summary, our proposed diffusion-based method for decoding the latent space of compression models offers significant improvements in audio quality compared to standard decoders. While it does require more compute and is slower, our results demonstrate that the trade-off is well worth it. Our approach generates audio that is more natural and in distribution, with fewer artefacts compared to existing methods. However, it is worth noting that our method may not be suitable for all use cases. For instance, if real-time performance is a critical factor, our approach may not be ideal.

**Ethical concerns.** Our approach, although not categorized as generative AI, can seamlessly integrate with techniques like Wang et al. [2023] to enhance the authenticity of generated voices. This advancement opens up potential missuses such as creating remarkably realistic deep fakes and voice phishing. Similar to all deep learning algorithms, our method depends on the quality and quantity of training data. We meticulously train our model on a substantial dataset to optimize its performance across a wide range of scenarios. Nevertheless, we acknowledge that imbalances in the dataset can potentially introduce biases that may impact minority groups.

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
