# OpenReview forum: "From Discrete Tokens to High-Fidelity Audio Using Multi-Band Diffusion"
_NeurIPS.cc/2023/Conference — NeurIPS 2023 poster_

### Official Review · Reviewer_tTLs · 2023-07-04

**Soundness:** 3 good
**Presentation:** 3 good
**Contribution:** 3 good
**Rating:** 5
**Confidence:** 4

**Summary:**

The paper proposes to use diffusion model to recover high-fidelity audio from compressed audio tokens. To improve the performance of diffusion model, the paper proposes multi-band diffusion with 1) Frequency Eq. Processor; 2) Scheduler Tuning; and 3) Band-Specific Training. The experiments show that the proposed method surpasses baseline method (EnCodec) by a margin.

**Strengths:**

1. The paper proposes a diffusion-based model as decoder for audio compression model (EnCodec), which surpasses the default decoder joint trained with encoder.

2. The paper proposes multi-band diffusion model to improve the generation quality.

**Weaknesses:**

1. The motivation of multiband is not clear. According to the description in paper, the multi-band is conducted on the hidden of encodec, which seems not have frequency information. If there is no intuition or explanation about the multi-band, it seems that using multi-band is a kind of improving the model parameters (using multiple model, one per band).

2. The experiments are only on EnCodec tokens, which are insufficient. The design in paper should be verified on vocoder tasks (recovering audio from mel-spectrum) and compared with more baselines (e.g., bigvgan, hifigan, wavegrad and multi-band MelGAN). Since mel-spectrum has frequency information, using multi-band on mel-spectrum is more convincing.

3. In ablation study, the comparison with single band seems to be unfair in terms of total model parameters.

**Questions:**

My main concern is the proposed method should be verified on vocoder task and compared with GAN based and diffusion based vocoder baselines.

---

> ### Author Rebuttal · Authors · 2023-08-09
>
> Thank you for your review and suggestions.
>
> **Clarification about the setting:** We will clarify the model's input-output relationship in the paper, as it currently appears to be unclear. All models take the full Encoder compressed representation as input. The diffusion process occurs in the waveform domain, specifically on waveforms that have undergone filtering using a passband filter.
>
> **Multi Band motivation:** We use a multi band approach is to address the issue of entangled errors that arise when conducting diffusion on full-band audio. In our experiments, this phenomenon manifested regardless of the model sizes. Our intuition suggests that the model overly relies on lower frequencies already present in the noisy waveform during training steps with small t (close to the clean x_0). Training independently for different frequency bands resolves this problem, preventing the model from inferring high-frequency content from the low frequency of noisy waveforms and requiring it to extract such information from the conditioning.
>
> **Discrete Units Focus:** Our paper is focused on generating from low bitrate discrete representation. Everything has been designed with this task in mind to seamlessly plug in prior text to audio work. Performing mel spectrogram to audio is not a task we considered yet. However we’ll add some discrete version of HifiGan and PriorGrad as additional baselines to our work. We’ll also add DAC(https://arxiv.org/abs/2306.06546), a new SOTA Compression model from July 2023. At 6kbps using their public implementation we have comparable performances. However we want to stress that this is not a decoder based on EnCodec but a completely different compression model. It is possible that replacing their decoder with diffusion would improve performances, which is something we might investigate in the future.
>
>
>
> |                               | Mean              | CI95       |
> |--------------------------|--------------------|-------------|
> | Ground Truth       | 90.32              | 1.39        |
> | MBD                      | *85.16*           |  0.93       |
> | Encodec               | 82.73              | 1.11        |
> | PriorGrad             | 65.16              | 2.2          |
> | HifiGan                 | 82.5                | 1.25        |
> |--------------------------|--------------------|-------------|
> | DAC                      | 84.44              | 1.14        |
> | OPUS                    | 65                   | 2.43       |
>
>
> **Comparison to Single Band:** We trained multi band models with 4 times less parameters to compare with the single band model from the ablation table (Table 3). We will add the line "Multi Band small" to the paper in table 3:
>
> | Setup | ViSQOL |  Mel SNR-L| Mel SNR-M | Mel SNR-H | Mel SNR-A |
> |--------|-----------|--------------|---------------|--------------|---------------|
> |Multi Band | 3.67±0.02 | 13.33 | 9.85 | 9.26 | 10.81|
> |Multi Band small | 3.56±0.03 |12.93| 9.81 | 9.11 |10.61 |
> |Single Band |3.32±0.02 |12.76| 9.82| 8.58| 10.39|

---

> > ### Comment · Reviewer_tTLs · 2023-08-16
> > **Thanks for your response**
> >
> > The further results address most of my concerns. I have increased my score to 5.

---

### Official Review · Reviewer_SCw1 · 2023-07-07

**Soundness:** 3 good
**Presentation:** 2 fair
**Contribution:** 2 fair
**Rating:** 6
**Confidence:** 2

**Summary:**

This paper presents Multi-Band Diffusion, a diffusion-based decoder for neural audio codecs. The proposed method demonstrates superior generation quality across various audio domains compared to a publicly available state-of-the-art neural audio codec method. Based on the analysis of time-frequency representation of audio, the model provides a novel diffusion-based model for audio synthesis.

**Strengths:**

* The authors provide extensive experimental results, along with a detailed explanation of their methodology.
* They present novel methods such as the band-specific diffusion model, frequency equalizer processor, and power noise scheduler, demonstrating their necessity and assessing their effectiveness through an ablation study.
* The model not only outperforms benchmarks in terms of the generation quality of reconstructed samples across different audio domains, but also proves its applicability in tasks such as text-to-audio and text-to-speech through empirical evidence.

**Weaknesses:**

* There is no comparison of generation quality when producing from high bit-rate latent representations. While the generation quality remains inferior compared to the ground truth audio, the authors only showcase generation quality from low bit-rate representations.
* There is a lack of a comparison of computational cost or model size. Although the authors mention that the proposed method requires more computation compared to standard decoders, a comparison of generation speed or parameter size would have provided a better understanding of the trade-off between generation quality and synthesis speed of this model. Furthermore, given that the model was trained using 1000 diffusion steps and only 20 steps were employed during inference, an evaluation of quality at different diffusion steps during inference could also be provided.

**Questions:**

From the selection of low bit-rate latent representation as the model's conditional input and the use of a small number of diffusion steps during inference, it seems that the authors have considered efficiency alongside the model's performance while empirically validating the proposed method. However, it would be beneficial to extend it and demonstrate the peak performance of the proposed model with larger bit-rate representation and high diffusion steps, as diffusion-based generative models have shown remarkable performance in various fields.

**Limitations:**

The authors have adequately addressed both the limitations of their research and its possible societal impacts.

---

> ### Author Rebuttal · Authors · 2023-08-09
>
> **High bit rate:** The Neural audio codec literature has mainly focussed on low bit rates. Moreover, these low bitrate setups align with the configurations employed by language modelling approaches that our method seeks to enhance, such as Musicgen, or Vall-E.. While our method is applicable to higher bit rates, the potential improvements are more marginal and might not justify the computational trade off. Finally, there are fewer use cases of high bit rate representations in the realm of text-to-audio (to the best of our knowledge, none).
>
> **Number of denoising steps:** We provide in the appendix of the paper a comparison of different number of denoising steps showing that there are very few improvement going more that 20 steps (see Table A5)
>
> **Computational time & Model size:** We will add this table to the appendix of the paper. Showing generation time and model size of EnCodec vs MBD also put the cost of a full pipeline LM + Decoder to put it in perspective.
>
> |                                                | Compute time (30s) | #parameters |
> |----------------------------------------|------------------------------|-------------------|
> | Encodec                                | 0.1s                            | 56M               |
> | MBD                                       | 21.2s                          | 411M             |
> | MusicGen-large + Encodec | 102s                           | 3.3B              |
> | MusicGen-large + MBD        | 123s                           | 3.7B              |

---

> > ### Comment · Reviewer_SCw1 · 2023-08-16
> > **Rebuttal Response**
> >
> > I thank the authors for addressing the previous concerns and emphasizing the importance of the model's performance at low bitrates, and presenting new evaluation results coupled with computational trade-offs. However, considering the results indicating this model as a high-performing, yet slower and larger alternative to EnCodec, I have chosen to maintain my initial review.

---

### Official Review · Reviewer_vmZN · 2023-07-07

**Soundness:** 3 good
**Presentation:** 3 good
**Contribution:** 2 fair
**Rating:** 5
**Confidence:** 4

**Summary:**

This paper proposes a novel multi-band diffusion (MBD) model that generates high-fidelity audio of multiple modalities, e.g., speech, music, and environmental sounds, from low-bitrate discrete representations. The authors show that MBD outperforms EnCodec and Opus in terms of perceptual quality.

**Strengths:**

The band-specific diffusion model is novel and appears to be a general technique for universal audio synthesis. The authors design a frequency equalizer to reduce the discrepancy between the prior Gaussian distribution and the data distribution in different frequency bands, which is sensible for improving the consistency and stability of audio generation, especially for general audios. The authors also propose a novel power noise scheduler, which empirically surpasses other commonly used schedulers, e.g., linear and cosine schedulers. I appreciate the high variety of audio generation experiments conducted for evaluating MBD. The generated samples of MBD on the demo page seem promising and are apparently better than EnCodec and Opus at the same bit rates.

**Weaknesses:**

1. Baselines are not strong enough. The authors only compare MBD to Encodec and Opus, where Opus is an old method proposed in 2012. I am therefore skeptical of the choice of the baselines in this paper. Are these baselines strong enough? The authors mention SoundStream (Zeghidour et al., 2021) but do not take it as a baseline. Considering the wide and successful application of SoundStream in many recent audio generation works, e.g., in AudioLM, Vall-E, and Natural Speech 2, it would be more convincing if MBD can be compared to SoundStream and evidently demonstrate superiority. I consider increasing my rating if such an experiment can be supplemented.

2. MBD depends on the frozen latent representations of a pre-trained EnCodec. The dependency of MBD on EnCodec complicates the analysis of the significance of this work. It remains unclear whether the high quality comes from a well-trained EnCodec or the proposed MDB model. The training of MBD also becomes more unstable. Please explain why the encoder of EnCodec cannot be jointly trained with MBD.

3. It is unfair that the generation speed and model size of MBD are not compared to those of Encodec. While the authors only present the strength of their proposed MBD model in terms of different quality measures, they do not mention the drawback of the generative diffusion model. As an iterative sampling method, it is foreseeable that MBD could be slow than Encodec up to orders of magnitude. It is then unfair to only focus on the quality when it is more reasonable to evaluate the audio codecs based on their practical values.

**Questions:**

Some questions have been stated above. In summary:

(i) What are the considerations when choosing the baselines? Are these baselines strong enough?

(ii) Is the frozen latent representations of a pre-trained EnCodec necessary for training MBD? Can MBD be end-to-end trained from scratch?

(iii) What are the speed and model sizes of MBD in comparison to EnCodec or other baselines (if any)?

**Limitations:**

The limitations of this work have been stated in the weaknesses.

---

> ### Author Rebuttal · Authors · 2023-08-09
>
> Thank you for your review and comments.
>
> **Comparison to SoundStream**: Despite our willingness to perform a direct comparison with SoundStream, it's important to note that the authors have not released a public repository, and there is no public reimplementation on GitHub that is reproducing SoundStream performances.
>
> **Baselines** EnCodec is a very strong Neural audio codec that builds on SoundStream. In their paper they provide a comparison showing that it outperforms SoundStream (cf. https://arxiv.org/pdf/2210.13438.pdf Appendix Table A2). Many models such as VALL-E or MusicGen are based on EnCodec. Natural speech 2 uses its own compression model that also builds on SoundStream. To add stronger baseline we add recent SOTA Compression Model DAC(https://arxiv.org/abs/2306.06546) and two other decoders (HifiGan and PriorGrad). We are comparable to DAC while building on different Encoders. It is likely that using MultiBand Diffusion as a decoder replacement for DAC improves performances over their GAN based decoder.
>
> |                               | Mean              | CI95       |
> |--------------------------|--------------------|-------------|
> | Ground Truth       | 90.32              | 1.39        |
> | MBD                      | **85.16**           |  0.93       |
> | Encodec               | 82.73              | 1.11        |
> | PriorGrad             | 65.16              | 2.2          |
> | HifiGan                 | 82.5                | 1.25        |
> |--------------------------|--------------------|-------------|
> | DAC                      | 84.44              | 1.14        |
> | OPUS                    | 65                   | 2.43       |
>
>
>
> **Aim of the paper:** Our approach serves as a replacement for EnCodec's decoder. We will highlight this aspect more prominently in the paper, as it has proven unclear to several reviewers. This approach offers the advantage of flexibility and compatibility across various applications. In the context of Text-to-Audio generation, it provides a means to swiftly preview audio using the fast and lightweight default decoder, and the option to switch to MultiBand diffusion when a desirable sample is identified and higher quality is needed.
> In all tables, the encoder and RVQ are shared between EnCodec and our method, thereby rendering the performance disparity solely attributable to our proposed diffusion decoder. Everything remains identical except for this aspect.
>
>
> **Training end-to-end compression with MBD:** This is an excellent question. We conducted preliminary experiments in this direction. However, we observed that utilising solely the L2 loss on the waveform does not yield satisfactory latent representations when compared to loss specifically designed for compression such as features and spectrogram matching, as is the case in a standard neural audio codec. How to combine the diffusion objective with perceptual losses is a complex and open problem that is beyond the scope of this paper.
>
> **Compute time:** We will include this table to the appendix of the paper including compute time and number of parameters for the different methods. We also add the comparison of complete pipelines using MusicGen to put the increase in compute time and model size in perspective.
>
> |                                                | Compute time (30s) | #parameters |
> |----------------------------------------|------------------------------|-------------------|
> | Encodec                                | 0.1s                            | 56M               |
> | MBD                                       | 21.2s                          | 411M             |
> | MusicGen-large + Encodec | 102s                           | 3.3B              |
> | MusicGen-large + MBD        | 123s                           | 3.7B              |

---

> > ### Comment · Reviewer_vmZN · 2023-08-11
> > **Response to the authors**
> >
> > I appreciate the authors' efforts made for the detailed response. The new strong baselines appear to be more convincing than the previous ones. Besides, following from the clarification from the authors, I am convinced that MBD is practically valuable for improving EnCodec's generation from codes. Yet, it seems unfair to me to compare a 411M MBD to a 8x smaller EnCodec. The immediate question raised here would be, if we train a 8x larger EnCodec, could it perform even better (I guess a ~400M EnCodec can still run much faster than MBD). Overall, considering the practical values of MBD, I decide to increase my previous rating to 5.

---

> > > ### Author Response · Authors · 2023-08-19
> > > **About a larger EnCodec**
> > >
> > > We thank the reviewer for considering the supplementary results we provided an increasing their ratings.
> > >
> > > **About the limitation of comparing to a smaller EnCodec model:** while we do not have immediately an 411M EnCodec model to compare to, we would like to highlight that in the original EnCodec paper, the authors tested a larger EnCodec model, in Table A.3 of the supplementary material of [Defossez et al. 2022]. In particular, the authors compared an EnCodec model with 48 initial hidden channels (the default with 56M parameters), to one with 64 initial hidden channels (which would have roughly 100M parameters). This study shows very limited changes in SI-SNR (going from 6.67 to 6.70 dB) and ViSQOL (4.35 to 4.38) when doubling the model size. This hints to the fact that the limitation of the EnCodec approach does not come from the model size, but instead from the adversarial training procedure.

---

### Official Review · Reviewer_s9xT · 2023-07-13

**Soundness:** 3 good
**Presentation:** 3 good
**Contribution:** 3 good
**Rating:** 7
**Confidence:** 4

**Summary:**

This is an interesting submission proposing the use of a diffusion model in a band-by-band manner to generate high-fidelity audio from (potentially) a variety of low bit rate inputs. The literature so far has only applied diffusion models for audio generation given spectrogram inputs. A specific diffusion noise schedule is proposed; and a diffusion-oriented frequency equalization method is proposed. Significant improvements in audio quality are described from a number of experimental evaluations.

**Strengths:**

Strengths:
- Clear presentation (in spite of persistent minor problems with grammar & spelling);
- Original multi-band diffusion model;
- Good literature review;
- Original frequency equalization model;
- Good empirical results, i.e. nice improvements in generated audio quality.



**Weaknesses:**

The main weakness is a persistent pattern of small mistakes in grammar and spelling. I mention a number below but this is just a small sample. Overall the work is very readable, but the problems with the writing should be corrected.

I note that there is mention of the proposed method being significantly slower than the alternatives examined, but I don't see a clear description of the compute cost of the proposed method compared to existing methods.

Specific comments:

> 27 artefacts [here and below]
--> artifacts

> have led to rich contextual representations that contains more

contains --> contain

> 32 They are optimized using complex combination
combination --> combinations

> 47 Results suggest the proposed method achieves significantly
> 48 superior performance than the evaluated baselines.

"significantly superior" is an odd phrasing, rewrite?

> 67 Clustering using a few centroids leads to the speech content representation being
> 68 mostly disentangled from the speaker and the f0 and thus controllable speech generation.

End of sentence is not grammatical; missing words before/around "thus controllable speech generation"?

"Eq. Processor" in Figure 1 caption vs. "EQ Processor" in the figure itself: choose one or the other, and keep consistent.

> 165 As a result, training a diffusion model on full-band audio data would always
> 166 provide the ground truth low frequencies when generating high frequencies

I don't quite understand this statement, rephrase?

>  170 Interestingly, dividing the frequency band
> 171 along model channels

I don't understand this statement. What are "model channels"?

> 238 5.1 Multi modalities model
Reading this section, i'm not sure what are the "modalities" referred to.

Table 1: "MBD", though it is clear that this is the proposed method, I don't see the acronym introduced. Introduce it early on, e.g. state "... Multi Band Diffusion (MBD) ..."?







**Questions:**

What is the computational cost of the proposed MBD compared to Opus and EnCodec?

**Limitations:**

The authors cite reasonable limitations to their work, including the possibility of their method being used to generate deep fakes, as well as the usual potential issue with training set bias, in spite of their efforts to avoid that.

---

> ### Author Rebuttal · Authors · 2023-08-09
>
> First thank you very much for the review and taking the time to point at typos and grammar issues. We have addressed all of your comments and will conduct a thorough proofreading of the paper to ensure that no error appears in the camera-ready version.
>
> **Regarding the computational cost:** We will include a table with the number of parameters and the compute time of our method vs Encodec, we also include in the total cost of a complete pipeline LM + Decoder such as the one described in the paper. This table will be included to the Appendix of the paper.
>
> |                                                | Compute time (30s) | #parameters |
> |----------------------------------------|------------------------------|-------------------|
> | Encodec                                | 0.1s                            | 56M               |
> | MBD                                       | 21.2s                          | 411M             |
> | MusicGen-large + Encodec | 102s                           | 3.3B              |
> | MusicGen-large + MBD        | 123s                           | 3.7B              |

---

### Official Review · Reviewer_kZTC · 2023-07-26

**Soundness:** 3 good
**Presentation:** 3 good
**Contribution:** 3 good
**Rating:** 6
**Confidence:** 4

**Summary:**

This paper presents a novel approach to processing audio data by developing and implementing a band-specific diffusion model, a frequency equalizer processor, and a power noise scheduler. The band-specific diffusion model processes various frequency bands independently, thereby reducing the accumulation of entangled errors. The frequency equalizer (Eq.) processor helps to lessen the discrepancy between the Gaussian prior distribution and the actual data distribution across different frequency bands by balancing energy-level between gaussian noise and each frequency band. The power noise scheduler, specifically designed for audio data with high sampling rate, is another contribution. The authors conduct extensive evaluations to gauge the efficiency of their approach, using both objective metrics and human studies. The results show that the proposed approach surpasses current state-of-the-art methods, encompassing both GAN and diffusion-based methods.

**Strengths:**

Generating high-frequency bands using diffusion model is something that has not been tackled very well. I personally have also suffered from this in various experiments.

All three contributions, training of band-specific models, frequency energy balancer, and power noise scheduler, are all valid approaches and were validated by metrics in experiments.

**Weaknesses:**

It would have been nicer if there were more GAN-based / Diffusion-based vocoder baseline models other than just encodec.

**Questions:**

1. Regarding the power noise schedule, what were the rules of thumbs to select hyperparameters?
2. Did the authors set the power noise schedule differently for each band?

**Limitations:**

The authors have addressed the limitations of this work in the conclusions such as slow speed of the current model.

---

> ### Author Rebuttal · Authors · 2023-08-09
>
> Thank you very much for your review.
>
> **Baselines:** We conducted a new subjective studies (MUSHRA) including new baselines such as a discrete version of HifiGan used in https://arxiv.org/abs/2104.00355  and Priorgrad (https://arxiv.org/abs/2106.06406) both conditioned on EnCodec units at 6 kbps. We also include DAC@6kbps(https://arxiv.org/abs/2306.06546), a concurrent SOTA neural compression model from July 2023. We underline that DAC is not 1-1 comparison since it isn't based on EnCodec codebooks but on its own learned latent space, it is likely that using MultiBand Diffusion on top of those codebooks would improve quality even more.
>
> |                               | Mean              | CI95       |
> |--------------------------|--------------------|-------------|
> | Ground Truth       | 90.32              | 1.39        |
> | MBD                      | *85.16*           |  0.93       |
> | Encodec               | 82.73              | 1.11        |
> | PriorGrad             | 65.16              | 2.2          |
> | HifiGan                 | 82.5                | 1.25        |
> |-|
> | DAC                      | 84.44              | 1.14        |
> | OPUS                    | 65                   | 2.43       |
>
>
> **Choice of Noise Schedule:** To determine our noise schedule function, we tested various functions, listened to diffusion process states, and empirically validated our choice and parameters through a grid search. Throughout this process, our primary objective was to have more steps in the less noisy region as described in the paper. We also adopted a schedule that wouldn’t include states where the model doesn’t manage to predict the noise more accurately than the Identity function.
>
> **Details about hyperparameters:** In order not to add unnecessary complexity, we used the identical noise schedule for every band, however we found that tuning the EQ processor differently for every band was beneficial. Comprehensive details will be added in the Appendix of the final version of the paper with all hyperparameters.

---

### Author Rebuttal · Authors · 2023-08-09

We want to make an ethical statement, the AudioGen and MusicGen team recently released the new public implementations with pre-trained models. We trained MultiBand Diffusion with those new versions of the compression models. We found improvement on MusicGen using MultiBand Diffusion as a decoder. However, in the first experiment that we did with pre-trained AudioGen, MBD was not improving over the baseline decoder. We want to come clean and say that we will remove the results on AudioGen as it is currently in the paper. However this doesn't change any other experiment conducted in the rest of the paper.

Results (MUSHRA) on MusicGen:

|                    | Mean  | CI95 |
|--------------------|-------|------|
| MusicGen + Encodec | 70.99 | 1.19 |
| MusicGen + MBD     | **74.97** | 1.94 |

---

### Comment · Area_Chair_sW4E · 2023-08-15

Dear reviewers,

The authors have uploaded their rebuttal.  Please take time to go over it.  If you have any further questions or concerns regarding the authors' rebuttal, please start a discussion.   If you are willing to adjust your scores after reading the rebuttal, please do.   For those who have already done it, thanks!

Best,

AC

---

### Decision · Program_Chairs · 2023-09-21

**Decision:**

Accept (poster)

**Comment:**

This paper proposes a so-called multi-band diffusion model that can produce high-fidelity signals of various audio modalities from low-bitrate discrete representations.  In this model, the diffusion is carried out in multiple independent bands. On top of that,  a frequency equalizer and a power noise scheduler are introduced to stabilize the training and improve the quality and efficiency of diffusion.  The authors compare the proposed multi-band diffusion model with existing high-performance models and show its superior quality  across various applications such as speech and music.   The work is considered novel and interesting.  The experimental results are solid.  The authors have cleared most of the concerns raised by the reviewers in their rebuttal and following discussion.  All reviewers are supportive to accept.